# Impact of PEG sensitization on the efficacy of PEG hydrogel-mediated tissue engineering

Alisa H. Isaac[1,2,3,7], Sarea Y. Recalde Phillips [1,7], Elizabeth Ruben[1], Matthew Estes[1], Varsha Rajavel[1], Talia Baig[1], Carol Paleti[4], Kirsten Landsgaard [1], Ryang Hwa Lee[4], Teja Guda [2,3], Michael F. Criscitiello [5], Carl Gregory [1,4] & Daniel L. Alge [1,6] ✉

While poly(ethylene glycol) (PEG) hydrogels are generally regarded as biologically inert blank slates, concerns over PEG immunogenicity are growing, and the implications for tissue engineering are unknown. Here, we investigate these implications by immunizing mice against PEG to stimulate anti-PEG antibody production and evaluating bone defect regeneration after treatment with bone morphogenetic protein-2-loaded PEG hydrogels. Quantitative analysis reveals that PEG sensitization increases bone formation compared to naive controls, whereas histological analysis shows that PEG sensitization induces an abnormally porous bone morphology at the defect site, particularly in males. Furthermore, immune cell recruitment is higher in PEG-sensitized mice administered the PEG-based treatment than their naive counterparts. Interestingly, naive controls that were administered a PEG-based treatment also develop anti-PEG antibodies. Sex differences in bone formation and immune cell recruitment are also apparent. Overall, these findings indicate that anti-PEG immune responses can impact tissue engineering efficacy and highlight the need for further investigation.

Hydrogels are versatile materials for medical device development, drug delivery, and tissue engineering due to their injectability, high water content, and soft-tissue like mechanical properties. Poly(ethylene glycol) (PEG) based hydrogels in particular are among the most popular synthetic biomaterials. Due to their synthetic nature, PEG hydrogels are highly tunable (mechanically and architecturally) and reproducible[1], unlike natural hydrogels made from polysaccharides (e.g., hyaluronic acid, dextran) and proteins (e.g., collagen, gelatin). Generally, PEG has hydroxyl end groups that can be converted to a multitude of functional groups suitable for crosslinking and conjugation to biomolecules such as extracellular matrix mimetic peptides that promote cell adhesion and enzymatic degradation. Peptide functionalized PEG hydrogels are widely used in tissue engineering research[2–5]. However, PEG hydrogels are also used for other applications. For example, PEG hydrogels have been used in implantable biosensors[6,7] and tissue sealants[8,9]. Several FDA-approved in situ crosslinking tissue sealants consist of PEG hydrogels, such as the ReSure Ocular Sealant from Ocular Therapeutix[10], the DuraSeal Exact Spine Sealant from Integra LifeSciences[11], and the Progel Pleural Sealant from Becton Dickenson[12]. Other FDA-approved PEG hydrogel-based implantable medical devices include the BioSentry Tract Sealant from AngioDynamics, which is used to seal lung biopsy tracts[13], the HydroMARK breast biopsy site marker[14], and the SpaceOAR device used in patients undergoing radiation treatment for prostate cancer[15].

[1]Department of Biomedical Engineering, Texas A&M University, College Station, TX, USA. [2]Department of Biomedical Engineering and Chemical Engineering, The University of Texas at San Antonio, San Antonio, TX, USA. [3]Department of Cell Systems and Anatomy, The University of Texas Health San Antonio, San Antonio, TX, USA. [4]Department of Cell Biology and Genetics, School of Medicine, Texas A&M University, College Station, TX, USA. [5]Comparative Immunogenetics Laboratory, Department of Veterinary Pathobiology, Texas A&M University, College Station, TX, USA. [6]Department of Materials Science and Engineering, Texas A&M University, College Station, TX, USA. [7]These authors contributed equally: Alisa H. Isaac, Sarea Y. Recalde Phillips. ✉e-mail: dalge@tamu.edu

PEG is widely regarded as bioinert due to its low toxicity, and the FDA considers PEG a generally recognized as safe additive and allows its use in products like cosmetics, toothpaste, and processed foods. Moreover, PEG hydrogels are often referred to as blank slates because of PEG's resistance to protein adsorption, which stems from its ability to bind water and its high chain mobility[16,17]. However, in recent years, concerns over PEG immunogenicity have grown and a surprisingly high prevalence of anti-PEG antibodies in the general public has been documented. Several studies have been conducted to evaluate anti-PEG antibody prevalence in the healthy population and results show that 23–72% of the population already has anti-PEG antibodies[18,19]. This high antibody prevalence is thought to be due to frequent exposure to PEG-containing everyday products that make PEG ubiquitous in our environment[20]. Chen et al. investigated anti-PEG antibody prevalence and trends among a sample size of 1504 Han Chinese individuals in Taiwan and found that 25.7% had detectable levels of anti-PEG IgG and 27% had detectable levels of anti-PEG IgM[21]. There was also a strong negative correlation between age and antibody prevalence, as 20% of women above age 50 years had anti-PEG antibodies while 60% of women younger than 50 years had anti-PEG antibodies. The group also discovered that the incidence of anti-PEG IgM prevalence was 20.2% in men and 32.0% in women, suggesting that gender plays a significant role in antibody prevalence. Lubich et al. reported an anti-PEG antibody prevalence of 23% in a cohort of 710 healthy human subjects in Austria. They subsequently studied a cohort of 600 healthy human subjects recruited from Austria and several cities in the US (Ammon (ID), Elkhart (IN), Fargo (ND), Lakeland (FL), and Laredo (TX))[18] to determine if geographical location has an effect on the prevalence of individuals with anti-PEG antibodies. Although there was some geographical variability, they found that 24% of the individuals in the second cohort had anti-PEG antibodies. The highest antibody prevalence was reported by Yang et al. who reported anti-PEG antibody prevalence of 72% in a sample size of 377 healthy individuals[19].

A significant risk associated with the prevalence of anti-PEG antibodies is the potential for anaphylactic reactions following exposure to PEG-containing products. Cases of anaphylaxis have been reported after ingestion of Colyte[22,23] and following administration of the Pfizer-BioNTech and Moderna COVID-19 vaccines[24]. These mRNA COVID-19 vaccines have also been shown to increase antibody levels in anti-PEG antibody prevalent individuals after vaccine administration[25]. Anti-PEG reactions have also been shown to negatively affect the efficacy of PEGylated drugs, which are attached to PEG to mitigate uptake by the mononuclear phagocyte system[26,27] and reduce renal clearance by increasing the drug's excluded volume[28]. Some PEGylated therapeutics include PEG-asparaginase (Oncaspar)[29], PEG-interferon-a2a (Pegasys)[30], PEG-anti vascular endothelial growth factor (Macugen)[31], and PEG-uricase (Kystexxa)[32]. Mima et al. reported that injecting PEGylated ovalbumin in mice caused an increase in anti-PEG IgM antibodies and a rapid clearance of a second dose of the same PEGylated protein[33]. This rapid clearance of a subsequent dose of PEGylated therapeutics has been extensively validated and is termed accelerated blood clearance. Clinically, it has been observed in PEGylated drugs such as Oncaspar and Krystexxa which have shown reduced therapeutic efficacy in patients that have PEG antibodies[34]. Moreover, the phase IIb and III clinical trials of Pegnivacogen, a PEGylated anticoagulant, were terminated due to allergic reactions and reduced efficacy in patients with anti-PEG antibodies[35]. Due to the potential adverse effects, the FDA has recommended that patients in clinical trials for PEGylated therapeutics be pre-screened for anti-PEG antibodies. However, the implications of anti-PEG immune reactions for PEG hydrogel-based implants and tissue engineering strategies have not previously been studied.

In this work, we evaluate the impact of PEG immunogenicity on tissue engineering efficacy, specifically in the context of bone tissue engineering. Importantly, we show that PEG sensitization increases bone formation and causes differences in bone morphology when calvarial defects in mice are treated with bone morphogenetic protein 2 (BMP-2) loaded PEG hydrogels, particularly in males. We also show sex-dependent differences in anti-PEG antibody production and immune cell recruitment to the defect site, indicating a divergent response that may contribute to the observed differences in bone formation. These findings highlight a need for further investigation on PEG immunogenicity in tissue engineering research.

## Results and discussion

An overview of our study is presented in Fig. 1. For our PEG hydrogel-mediated bone tissue engineering treatment, PEG-based microporous annealed particle (MAP) hydrogel scaffolds were implanted into murine calvarial bone defects. MAP hydrogels were chosen because they can be easily formed within tissue defects and possess an interconnected microporous structure that has been shown to result in superior tissue regeneration when compared to conventional, nanoporous PEG hydrogels[36,37]. The PEG hydrogel microparticles used for MAP scaffold fabrication that were synthesized via electrospraying (diameter = $531.88 \pm 100.44\ \mu m$; Supplementary Fig. 1) contained the cell-adhesive peptide CGRGDS and were crosslinked with the matrix metalloproteinase (MMP) degradable peptide KCGPQGIAGQCK (KCGPQ) (Fig. 1A). Prior to surgically creating calvarial defects and implanting PEG-based MAP hydrogels, the mice were subjected to a 4-week PEG immunization protocol, which consisted of weekly subcutaneous injections of multi-PEGylated keyhole limpet hemocyanin (PEG-KLH) (Fig. 1B). We used this immunization protocol because PEG-KLH injections were previously shown to be highly effective at inducing anti-PEG antibody production and resulted in higher titers compared to other PEGylated proteins[38]. Following immunization, plasma was collected to characterize the production of anti-PEG antibodies. Two weeks later, 2.7 mm calvarial defects were created and either left empty, treated with a PEG MAP hydrogel, or treated with a PEG MAP hydrogel containing a therapeutic 500 ng dose of BMP-2 (MAP + BMP-2). Three and 6 weeks post implantation, mice were humanely euthanized, plasma was collected, and the calvarial tissue was removed and analyzed for changes in bone growth and inflammation caused by the presence of anti-PEG antibodies.

Anti-PEG IgM and IgG titers ranging from 1:100 to 1:3200 were determined in all sensitized mice pre-implantation and at time of euthanasia via ELISA. Prior to surgery and hydrogel implantation, almost all mice sensitized with PEG-KLH were positive for anti-PEG antibodies. The exception was 1 female mouse that did not have a detectable titer equal to or greater than 1:100 (Supplementary Fig. 2). Although there was variability in detectable antibody titers among the mice, these results indicate successful induction of a robust anti-PEG immune response. Despite the presence of anti-PEG antibodies, no adverse effects were noted in the PEG sensitized mice following implantation of the PEG-based MAP scaffolds. Qualitative analysis by μCT revealed that BMP-2 delivery from the PEG-based hydrogels stimulated bone formation and partial regeneration of the calvarial defects (Fig. 2 and Supplementary Fig. 3). In contrast, the growth factor-free hydrogels did not result in increased bone formation compared to the untreated, empty defect controls. These results were expected based on the well-known osteoinductivity of BMP-2 and due to the lack of other osteoinductive cues in the MAP hydrogels.

Interestingly, bone volume measurements from the μCT analysis revealed that the increase in bone formation observed with PEG hydrogel-mediated BMP-2 delivery was generally greater in the PEG sensitized mice compared to naive mice of the same treatment group (Fig. 2B). At 3 weeks, the volume of bone formed in sensitized male and female mice treated with the MAP + BMP-2 hydrogels was $1.63 \pm 0.56$ and $1.45 \pm 0.22$ mm³, a 1.71- and 1.44-fold increase in bone formation, respectively, when compared to their naive counterparts. Statistical analysis via a two-way ANOVA mixed effects model revealed that,

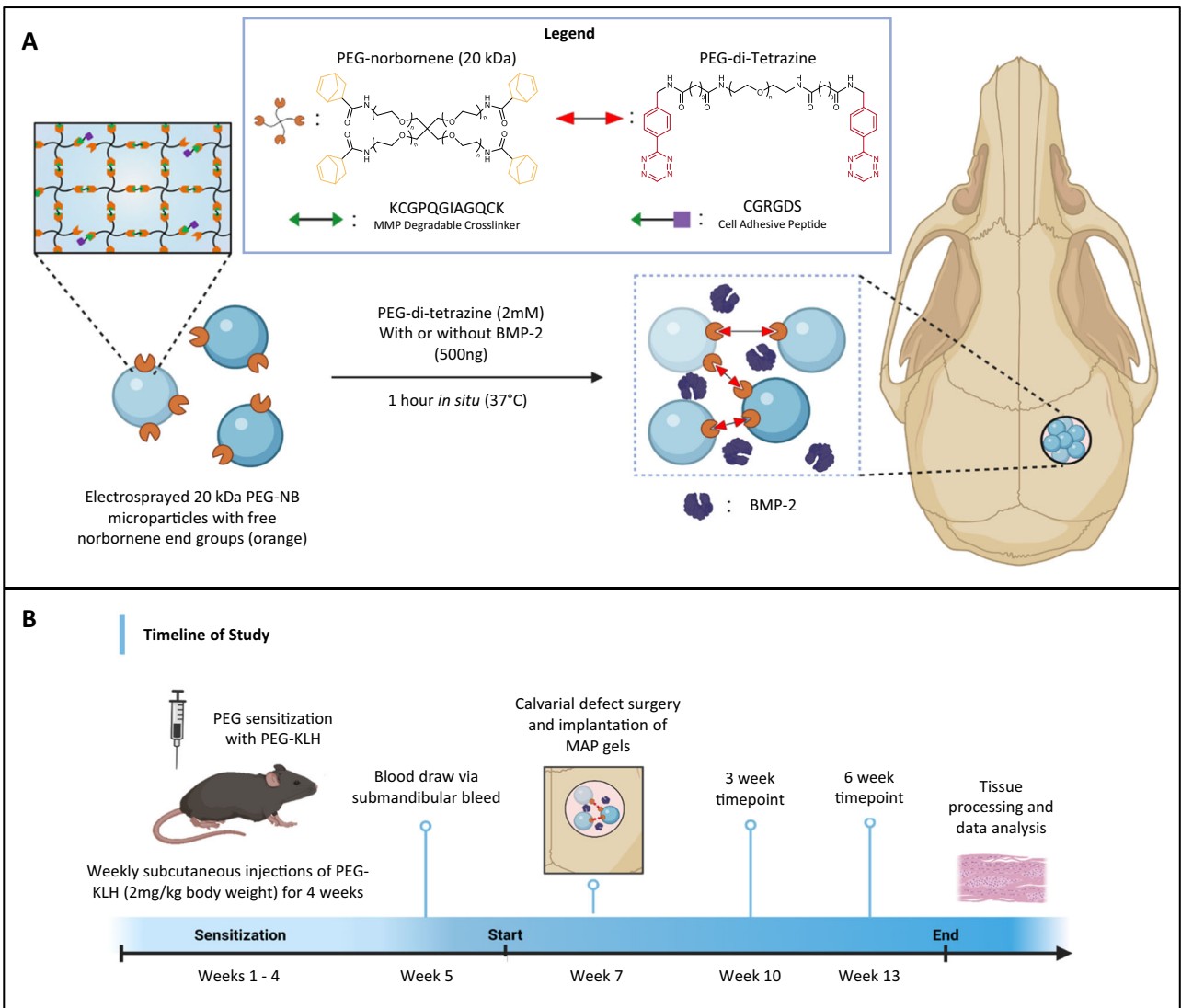

**Fig. 1 | The impact of PEG sensitization and anti-PEG antibodies on the efficacy of a PEG hydrogel-based tissue engineering strategy was assessed via implantation of PEG-based MAP hydrogels in a calvarial defect model.** Schematic of **A** materials and fabrication of PEG microparticles and annealing of MAP hydrogels, and **B** timeline of the animal study. The calvarial defect model was chosen because it is widely used to evaluate the efficacy of biomaterials and therapeutics for bone regeneration. Tissue samples were collected at 3 and 6 weeks post-implantation. Tissue processing and data analysis were conducted immediately following the end of each time point. Created with BioRender.com.

in addition to the significant effect of defect treatment type (i.e., empty defect, MAP, or MAP + BMP-2; $p < 0.01$ for both males and females), the interaction between PEG sensitization and treatment type was also statistically significant for both sexes ($p = 0.0095$ and $p = 0.042$ for males and females, respectively). Post hoc comparisons between sensitized and naive mice receiving the MAP + BMP-2 treatment confirmed a strong effect of PEG sensitization ($p = 0.0521$ and $p = 0.0663$ for males and females, respectively). Statistically significant increases in bone volume compared to the empty defect group and the MAP hydrogel treatment group at this early timepoint were also only found for the PEG-sensitized mice ($p < 0.005$ and $p < 0.0005$ for males and females, respectively).

At 6 weeks, BMP-2 continued to drive bone formation, as the MAP + BMP-2 groups had the highest volume of bone quantified at the defect site compared to the other treatment groups, regardless of sex or sensitization to PEG. The volume of bone formed in sensitized male and female mice treated with the MAP + BMP-2 hydrogels was $2.04 \pm 0.89$ and $1.62 \pm 0.30$ mm³, respectively, compared to $1.61 \pm 0.44$ and $1.33 \pm 0.69$ mm³ for naive male and female mice, respectively,

receiving the same treatment. Two-way ANOVA analyses on the 6-week data revealed that, in addition to the significant effect of defect treatment type ($p < 0.01$ for both males and females), sensitization had a strong effect in the males ($p = 0.0502$). This effect was noted despite having an outlier in the sensitized MAP + BMP-2 group that exhibited markedly lower bone formation compared to the other 3 replicates and was below 1.5× the interquartile range of the first quartile. Notably, this mouse experienced some minor bleeding from its incision, which may have impacted the efficacy of the treatment, and excluding this data point would have resulted in a statistically significant difference between sensitized and naive MAP + BMP-2 treated males. Nevertheless, even with the inclusion of the outlier, statistically significant differences in the post hoc comparisons were only seen when comparing the sensitized MAP + BMP-2 group to the naive empty defect and naive MAP only control treatments ($p < 0.05$ and 0.01, respectively), and the sensitized MAP + BMP-2 vs. sensitized MAP comparison was nearly significant ($p = 0.0574$). Significant differences were not noted in the comparisons involving the naive MAP + BMP-2 males, although the comparison to the naive MAP only control treatment was

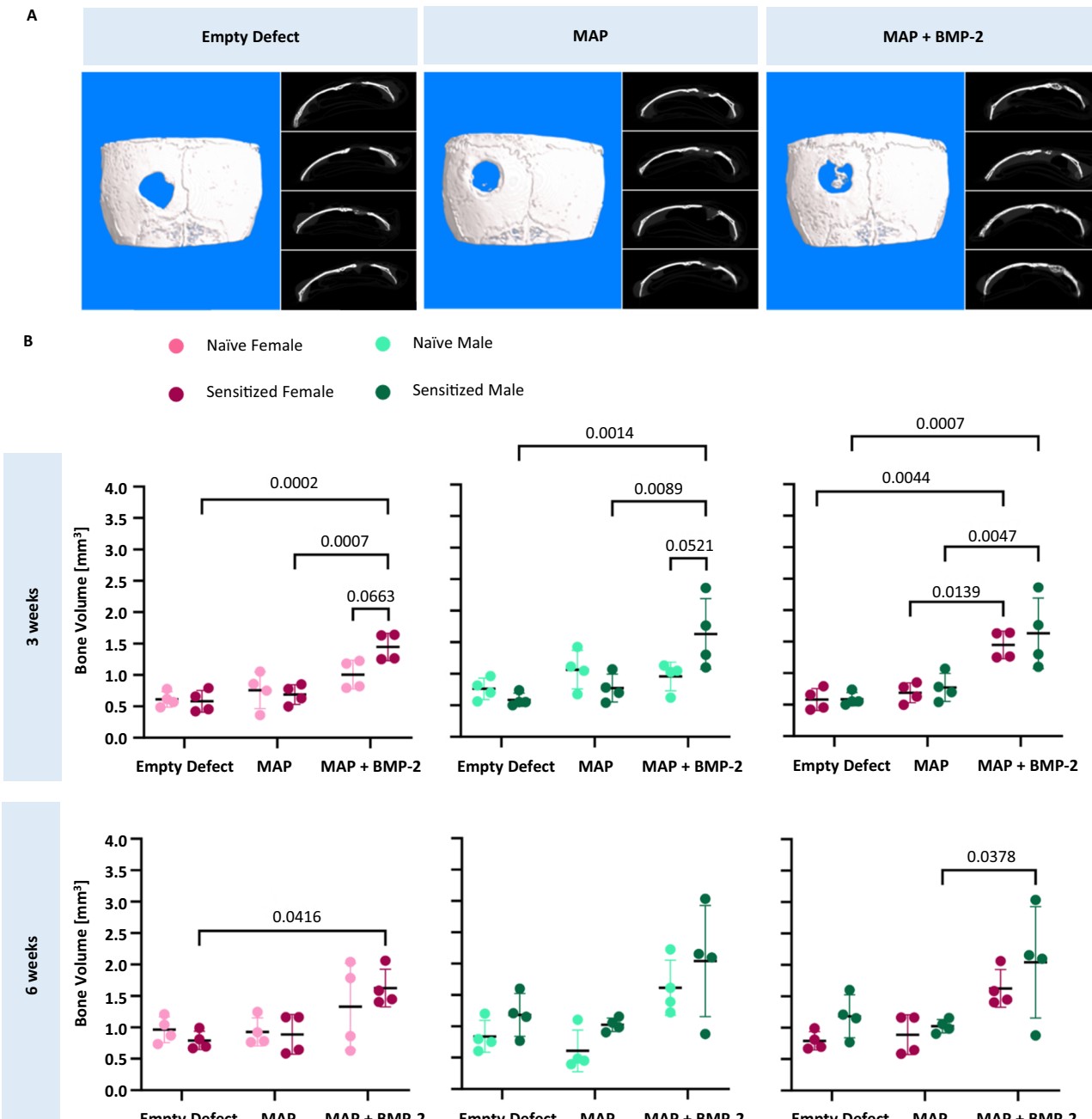

**Fig. 2 | PEG sensitization contributes to increased bone in PEG MAP + BMP-2 treatment groups. A** Representative 3D reconstructions of μCT scans and axial images of PEG sensitized male mice 6 weeks post-implantation. **B** Bone volume measurements from μCT scans of calvarial defects at 3 and 6 weeks post-implantation ($n = 4$ mice per treatment group). Statistical significance was calculated by ordinary two-way ANOVA with Tukey's multiple comparisons test with a single pooled variance, an alpha threshold of 0.05, and 95% confidence interval. Data are represented as mean ± SD, and dots represent individual specimens.

nearly significant ($p = 0.0611$). For the females, the only statistically significant comparison in the 6-week data was between the sensitized MAP + BMP-2 group and the sensitized empty defect control ($p < 0.05$). Sex also appeared to have an impact on bone formation among sensitized mice, as sensitized females generally had slightly reduced bone volume measurements with the MAP + BMP-2 treatment compared to their male counterparts (Fig. 2B).

Histological analysis provided further insight into the differences in bone regeneration in mice treated with BMP-2 loaded MAP hydrogels. Interestingly, calvarial tissue surrounding the defect sites exhibited morphological differences when compared to the uninjured contralateral side. In general, the parietal bone where the defects were created had a less dense and woven collagen architecture, and the tissue neighboring the defect sites was noticeably thicker than the tissue on the contralateral side (Fig. 3). Increased bone growth was apparent histologically in the PEG-sensitized mice (Figs. 3 and 4), which was consistent with the increased bone volume observed in the μCT data. In addition, the overall thickness of the new bone in the defects was greater in the PEG-sensitized mice (Figs. 3 and 4). Morphological changes in collagen structure between sensitized and naive mice were noted as well. Especially in males, naive mice were observed to have matured bone tissue as noted by dark and structured blue collagen staining from the Masson's trichrome, whereas their PEG-sensitized counterpart exhibited woven and less dense collagen staining (Fig. 4). Quantitative analysis of collagen deposition from the Masson's trichrome staining revealed that PEG-sensitized mice treated

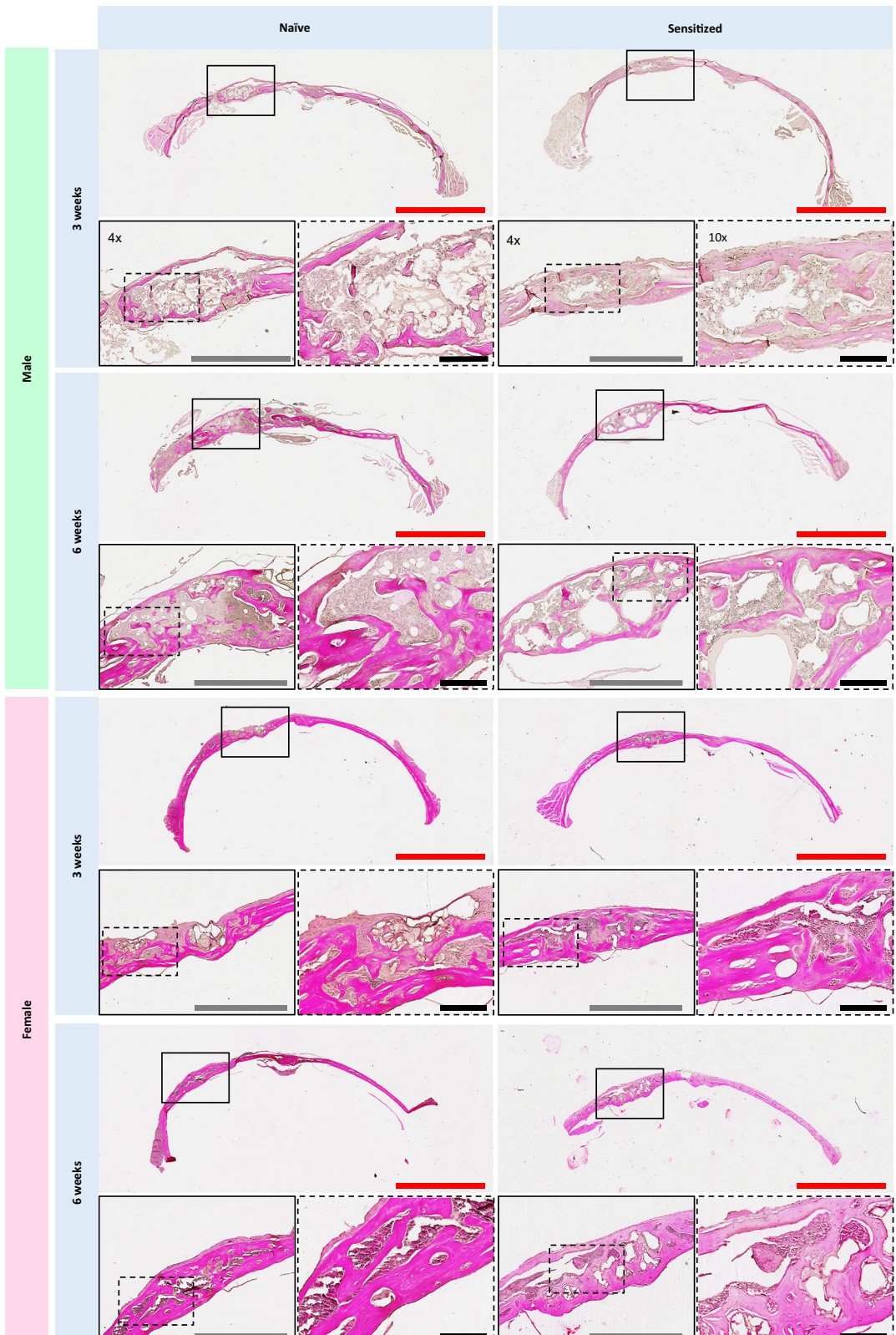

**Fig. 3 | PEG MAP + BMP-2 treatment groups demonstrate increased matrix deposition at the defect site and morphological changes to the surrounding tissue.** Images of H&E staining of calvarial defects of naive and sensitized top male and bottom female mice treated with PEG MAP + BMP-2 at 3 and 6 weeks post-implantation. Tissue samples from $n = 4$ mice per treatment group were analyzed. The entire calvaria was imaged to show the defect site (left side) and healthy, uninjured bone in the contralateral tissue (scale bar = 3 mm). Defect sites are indicated in solid-line boxes and magnified at ×4 (scale bar = 1 mm), and portions of the defect site that are magnified at ×10 are indicated in dashed-line boxes (scale bar = 200 μm).

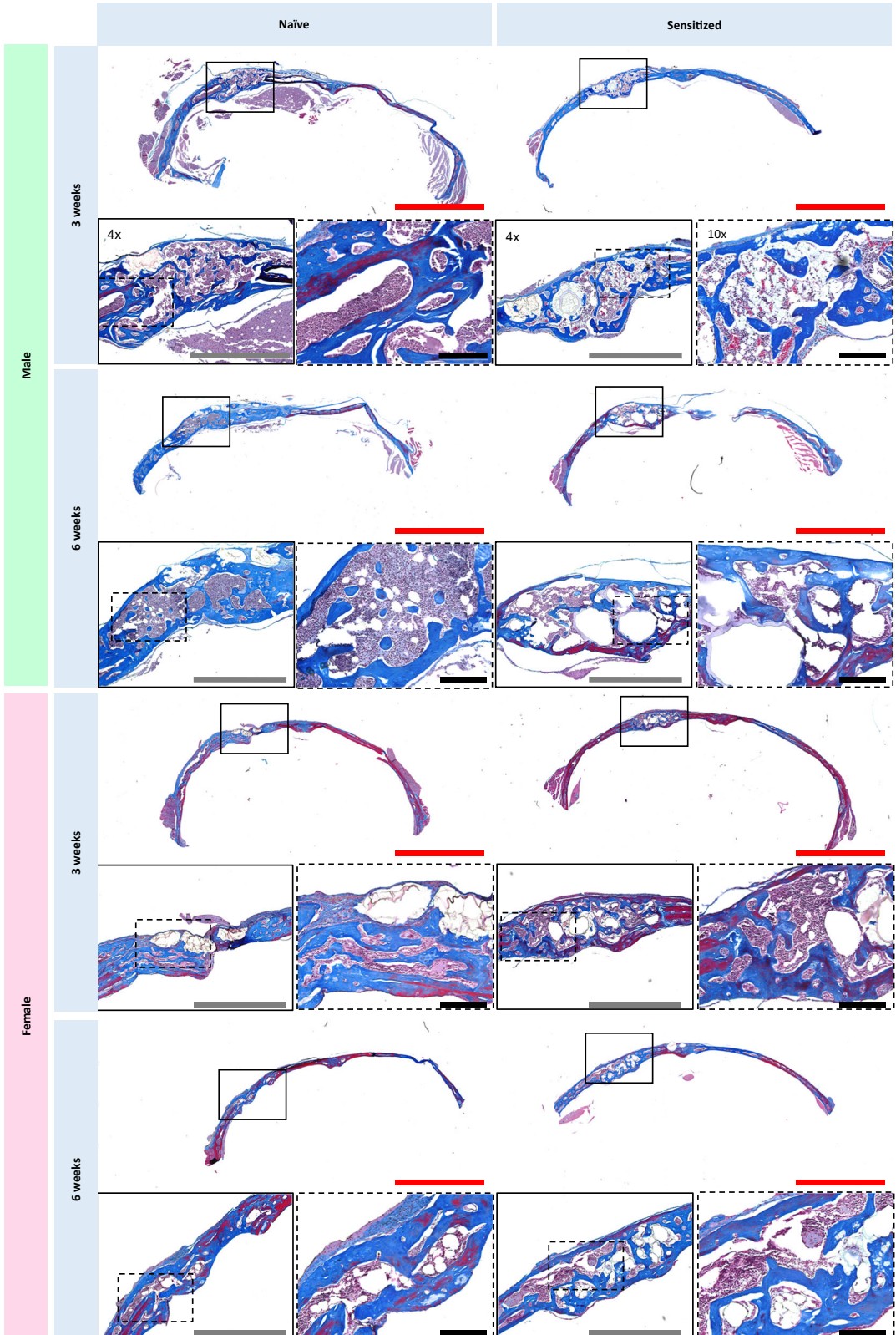

**Fig. 4 | PEG sensitization among PEG MAP + BMP-2 treatment groups contributed to disorganized collagen structure when compared to their naive counterparts.** Masson's Trichrome staining of calvarial defects of naive and sensitized top male and bottom female mice treated with PEG MAP + BMP-2 at 3 and 6 weeks post-implantation. Tissue samples from $n$ = 4 mice per treatment group were analyzed. The entire calvaria was imaged to show the defect site and healthy, uninjured bone in the contralateral tissue (scale bar = 3 mm). Defect sites are indicated in solid-line boxes and magnified at ×4 (scale bar = 1 mm), and portions of the defect site that are magnified at ×10 are indicated in dashed-line boxes (scale bar = 200 µm).

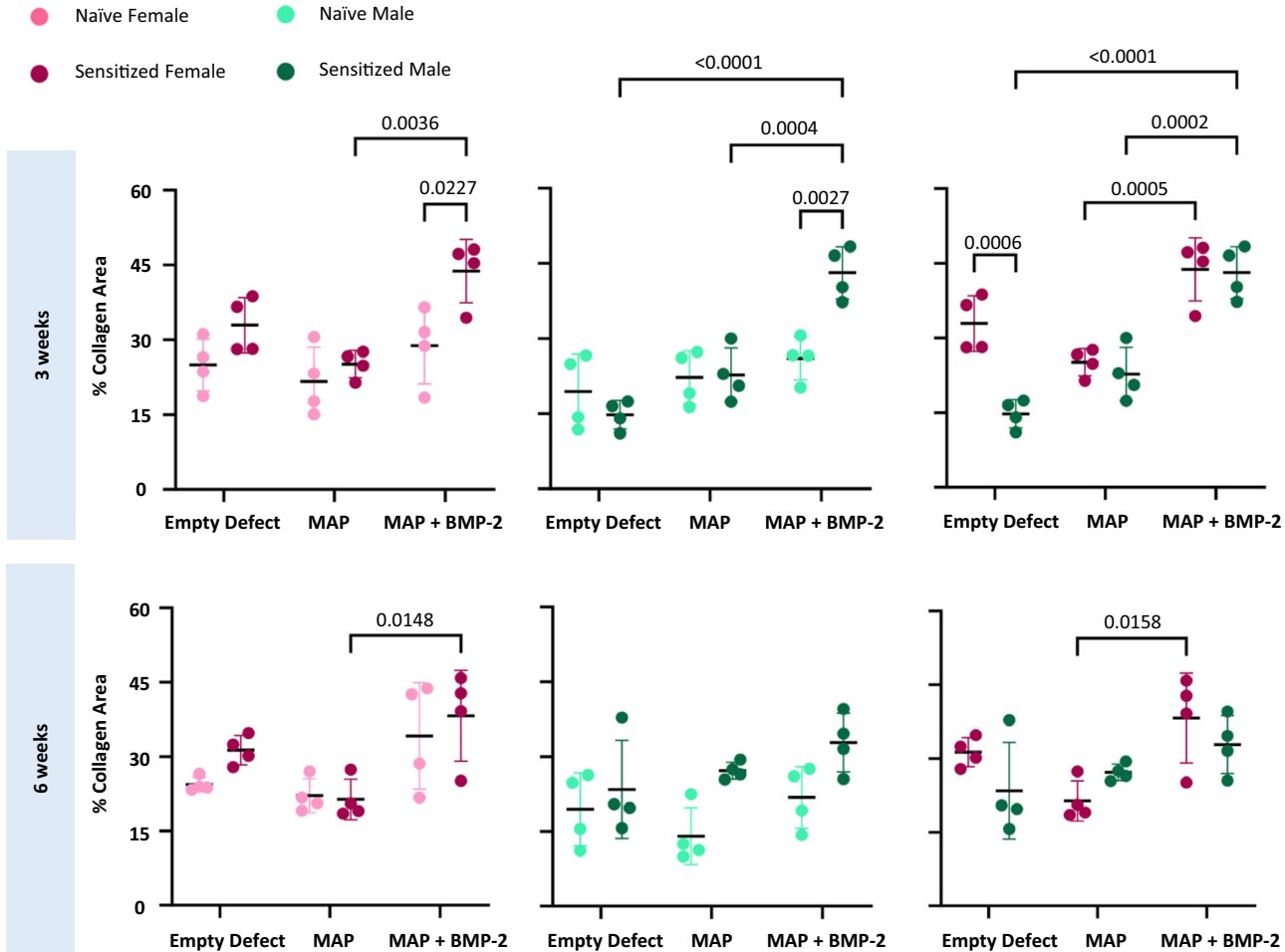

**Fig. 5 | Quantitative analysis of the matrix at defect sites supports that PEG sensitization contributes to increased collagen deposition.** Collagen deposition was determined from tissue samples subjected to Masson's Trichrome staining ($n = 4$ mice per treatment group). Statistical significance was calculated by ordinary two-way ANOVA with Tukey's multiple comparisons test with a single pooled variance, an alpha threshold of 0.05, and 95% confidence interval. Data are represented as mean ± SD, and dots represent individual specimens.

with MAP + BMP-2 demonstrated higher amounts of collagen deposition at the defect site than their naive counterparts, as well as the highest values of all treatment groups overall (Fig. 5). This increase was significant among the male (naive vs. sensitized: 26.0 ± 4.31% vs. 43.1 ± 5.22%) and female (naive vs. sensitized: 28.8 ± 7.64% vs. 43.8 ± 6.34%) MAP + BMP-2 groups at 3 weeks. Considering that collagen is the major component of the extracellular matrix of bone[39,40], these findings corroborate the μCT findings at 3 weeks.

Overall, the data suggest that PEG sensitization increases bone volume, increases bone thickness, and may cause bone remodeling to be at earlier stages in the calvarial defect compared to PEG naive mice of the same treatment groups. Moreover, in males specifically, the bone formed in the calvarial defects exhibited morphological differences due to PEG sensitization. Specifically, the architecture was more porous compared to their naive counterparts (Figs. 3 and 4). Large cavities could be attributed to the microgels comprising the MAP hydrogel, whereas small cavities interspersed in the trabecular bone could be areas of bone marrow production and adipocytes present in the newly formed bone matrix[41]. Additionally, porous morphology in newly formed bone is a characteristic of early-stage BMP-driven bone formation[42,43], and this high porosity could compromise the overall structural integrity of the bone formed at the defect site. The higher bone porosity observed in sensitized mice treated with MAP + BMP-2 could also suggest changes in MAP gel degradation due to changes in MMP secretion by infiltrating cells in sensitized mice. The implanted

MAP gels were engineered with an MMP-degradable crosslinker, KCGPQGIAGQCK, to allow for cell induced degradation of the implanted hydrogel as the bone regenerated. This peptide can be degraded by MMP-1, -2, -3, -7, -8, and -9[44]. Multiple cell types involved in the inflammatory and remodeling stages produce MMPs, including osteoclasts, macrophages, lymphocytes, and granulocytes[45,46]. Macrophages in particular secrete large amounts of MMP-2, which the MMP degradable crosslinker used in the MAP hydrogels is highly susceptible to degradation by[44,45]. Interestingly, male mice had a greater abundance of cells (Fig. 3) compared to females, which may have contributed to increased MMP production and MAP hydrogel degradation, and subsequently the increased presence of large empty cavities observed in sensitized males compared to females.

To elucidate what cells may be contributing to the observed differences, we investigated infiltration of CD68 and CD3 positive cells within the defects of mice treated with MAP + BMP-2. CD68 was of interest because macrophages and osteoclasts are positive for this marker. These cells are major contributors of MMP secretion, and an increase in the presence of these cells may increase the MAP hydrogel degradation rate and influence bone formation. Moreover, it is well documented that antibody binding, or opsonization, efficiently recruits macrophages through the release of chemotactic complement protein 5a (C5a), shortly following the activation of complement[47]. At 3 weeks, there was an increasing trend in the number of CD68+ cells compared to their naive counterparts, as well as between males

compared to females (Fig. 6A). However, no statistically significant differences were found. Significant differences in CD68+ cell presence were noted at 6 weeks. Specifically, the number of CD68+ cells was statistically higher in sensitized females compared to their naive counterparts ($p = 0.0168$). Additionally, sex differences were observed among the naive group, as males had a significantly higher presence of CD68+ cells compared to females ($p = 0.0163$). Naive male mice also exhibited a slight increase in CD68+ cells present at the defect site when compared to their sensitized counterparts, but this difference was not significant ($p = 0.1773$). The presence of CD3+cells was also investigated, as this is a cell surface marker represented on T cells. T cells play a key role within adaptive immunity and contribute to the activation of macrophages and cytotoxic T cells, and antibody class switching to IgG[48,49]. At 3 weeks, the effect of sensitization was highly significant ($p = 0.0001$), and the interaction parameter between sex and sensitization was significant ($p = 0.0367$). This result indicates increased involvement of adaptive immunity in the response to PEG hydrogels in sensitized mice, which can be explained by the fact that T cells are induced to proliferate by a peptide:MHC complex on antigen presenting cells, a cellular activity aided by antibody opsonization[50]. Sensitized females had a statistically significant increase of CD3+ cells within the defect compared to naive females ($p = 0.0006$) as well as naive males ($p = 0.0455$), but the difference between sensitized and naive males was not significant ($p = 0.1645$) (Fig. 6B). The differences between males and females of the same experimental groups were also not statistically significant at 3 weeks. However, at 6 weeks, sensitized males had a significantly lower number of CD3+ cells compared to naive males ($p = 0.0053$) as well as sensitized females ($p = 0.0146$). Prior studies have shown that T cells can promote osteoclastogenesis and CD8+ T cells in particular negatively regulate bone healing[51]. Thus, the decreased presence of CD3+ cells in sensitized males suggests that a divergent T cell response may have contributed to the observed differences in bone formation.

Both CD3 and CD68 positive cell presence may be closely tied to anti-PEG antibody prevalence in the MAP + BMP-2 groups. Two isotypes of anti-PEG antibodies of interest include IgM and IgG. The IgM isotype class is the first antibody isotype created by B cells after exposure to an antigen[52]. It has low-affinity, high-avidity, and efficiently activates complement[53,54]. IgG antibodies, on the other hand, are the most predominant isotype, highly specific, long-lasting, and due to their smaller size can leave the bloodstream and travel into tissues[55]. Investigating the detectable titers of both these isotypes in the sensitized mice during the duration of the study provided insight into the involvement of the adaptive immune response to the MAP + BMP-2 hydrogels, as sensitized males were found to generate more anti-PEG IgM and IgG antibodies than sensitized females prior to MAP hydrogel implantations (Supplementary Fig. 2). The trend of increased CD68+ cell presence observed at 3 weeks in the males may be related to the higher anti-PEG titers prior to implantation. Anti-PEG IgM antibodies continued to be detected at higher antibody titers in sensitized males than females post-implantation, but isotype switching to IgG was observed in 50% of the females treated with MAP + BMP-2 at 3 weeks post-implantation (Fig. 7A). The presence of IgG antibodies requires T cell-mediated antibody switching and may explain the significantly increased presence of CD3+ T cells in sensitized females compared to males. Although both sensitized males and females were treated with MAP + BMP-2 hydrogels, it is clear that the anti-PEG response is sex-dependent and affects the presence of inflammatory cells, antibody production, and ultimately PEG hydrogel mediated bone regeneration (Fig. 2A).

An unexpected observation was that naive male mice treated with MAP + BMP-2 displayed an increase in CD68 and CD3 positive cells at 6 weeks post-implantation. Based on this observation, we performed ELISAs on the plasma of naive mice that were collected at the time of euthanasia to test for the development of anti-PEG antibodies. Due to

material limitations, only three mice per treatment group were analyzed. Interestingly, all naive males treated with MAP + BMP-2 had developed anti-PEG IgM antibodies by week 3 (Fig. 7B). Females with the same treatment also developed IgM antibodies, but at much lower detectable titers. Some females, treated with either MAP or MAP + BMP-2, also developed IgG antibodies at 6 weeks, which was not evident in males, but once again at low titers. The high detectable titers, up to 1:6400, observed in the male mice treated with MAP + BMP-2 may be related to the high number of CD3 and CD68 positive cells at the site of implantation. Notably, this is the first study to demonstrate that PEG hydrogel-based tissue engineering scaffolds on their own can induce anti-PEG antibody formation in vivo. Future studies should investigate the implications of this finding.

The differences in bone growth observed between sensitized and naive mice, especially in males, also warrant further investigation. We hypothesize that these differences could be due to changes in inflammatory cytokine production, crosstalk between innate and adaptive immunity, and complement activation. Systemic cytokine production was assessed on plasma collected from males administered MAP + BMP-2 treatment, and negligible differences were observed between sensitized and naive groups (Supplementary Fig. 8). However, this does not negate the possibility that localized cytokine production was modulating bone formation at defect sites. Increased inflammatory cytokine secretion in PEG sensitized mice could be driven by opsonization of the hydrogels with anti-PEG antibodies, which would, in turn, attract immune cells such as macrophages to the MAP hydrogel. When activated, macrophages secrete inflammatory cytokines like TNF-α and IL-6 during the innate immune response that could contribute to the outcome of bone formation. TNF-α is a pro-inflammatory cytokine often associated with osteoclast differentiation, but in the presence of BMP-2 has been observed to promote bone formation compared to TNF-α on its own[56]. IL-6 is secreted by activated macrophages in response to injury[57,58] and plays a crucial role in osteogenic outcomes[59]. This cytokine aids in mediating the dynamic between bone formation and resorption, with evidence highlighting its specific contribution toward osteoclast differentiation through increased expression of receptor activator of nuclear factor-κB ligand (RANKL)[59,60]. IL-6 secretion has been observed to influence B cell differentiation and promote antibody production[61], with one study reporting that co-administration of IL-6 with an antigen promoted production of IgG-specific antibodies in mice[62]. If anti-PEG production is stimulated, it could influence bone formation and morphology and change osteogenic outcomes. Complement activation following hydrogel opsonization by anti-PEG antibodies could also play a role in the observed differences in bone formation. When the complement system is activated by opsonized surfaces, complement component 3 (C3), which is often secreted by osteoclasts[63], is cleaved into C3a, which has been reported to induce osteoblastogenesis[63]. This potential mechanism is particularly intriguing since complement activation is thought to play a key role in hypersensitivity reactions to PEGylated therapeutics[64]. For example, Kozma et al. reported that administering 2-K-methyl-PEGylated liposomes (Doxebo) to PEG-sensitized pigs resulted in elevated levels of the terminal membrane attack complex of complement, a rise in pulmonary arterial pressure, and fatality in 2–4 min[65]. Clinically, hypersensitivity to PEGylated drugs has been linked to complement activation such as with Doxil, a PEGylated doxorubicin liposome[64].

In addition to sensitization causing changes in defect regeneration, BMP-2's role in bone formation must also be considered. Inflammatory cytokines IL-1, IL-6, and IL-17 have been documented to influence BMP-2 signaling and ultimately result in increased bone formation[66–68]. If inflammatory cytokine production is amplified due to PEG sensitization, an unintended increase in bone formation could have significant implications for the use of PEG hydrogels in the delivery of osteoinductive therapeutics. As shown in the histological

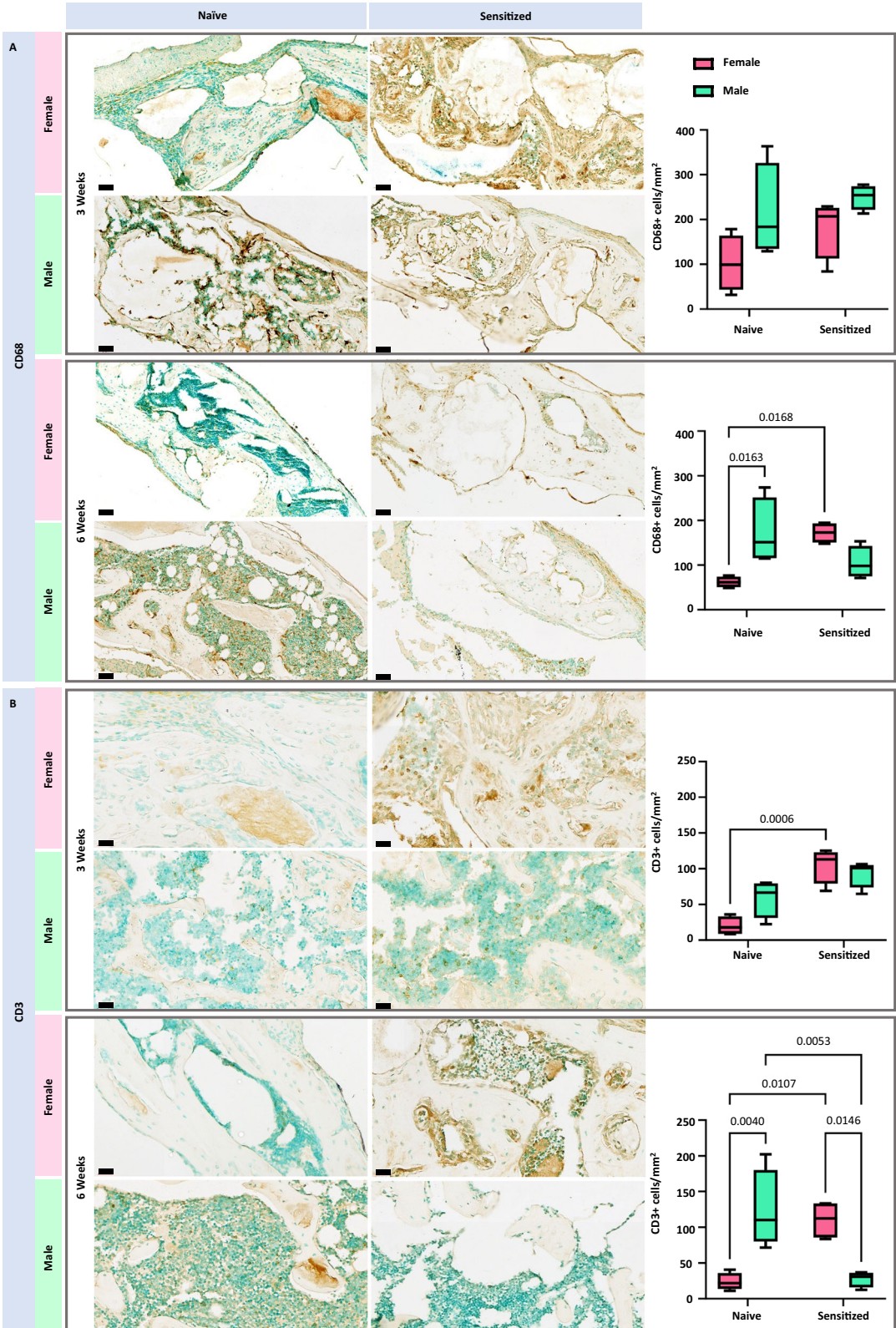

**Fig. 6 | Immune cell recruitment at defect sites administered PEG MAP + BMP-2 treatment was influenced by PEG sensitization at the 3- and 6-week timepoints, as well as by biological sex at the 6-week timepoint.** Quantification of **A** CD68+ cells and **B** CD3+ cells at defect sites 3 and 6 weeks post-implantation. Statistical significance was calculated by ordinary two-way ANOVA with Tukey's multiple comparisons test with a single pooled variance, an alpha threshold of 0.05, and 95% confidence interval. Center line = median; box limits = upper and lower quartiles; whiskers = 1.5× interquartile range ($n = 4$ mice per treatment group). Staining is representative of $n = 4$ sections per sample. Scale bar = 60 μm.

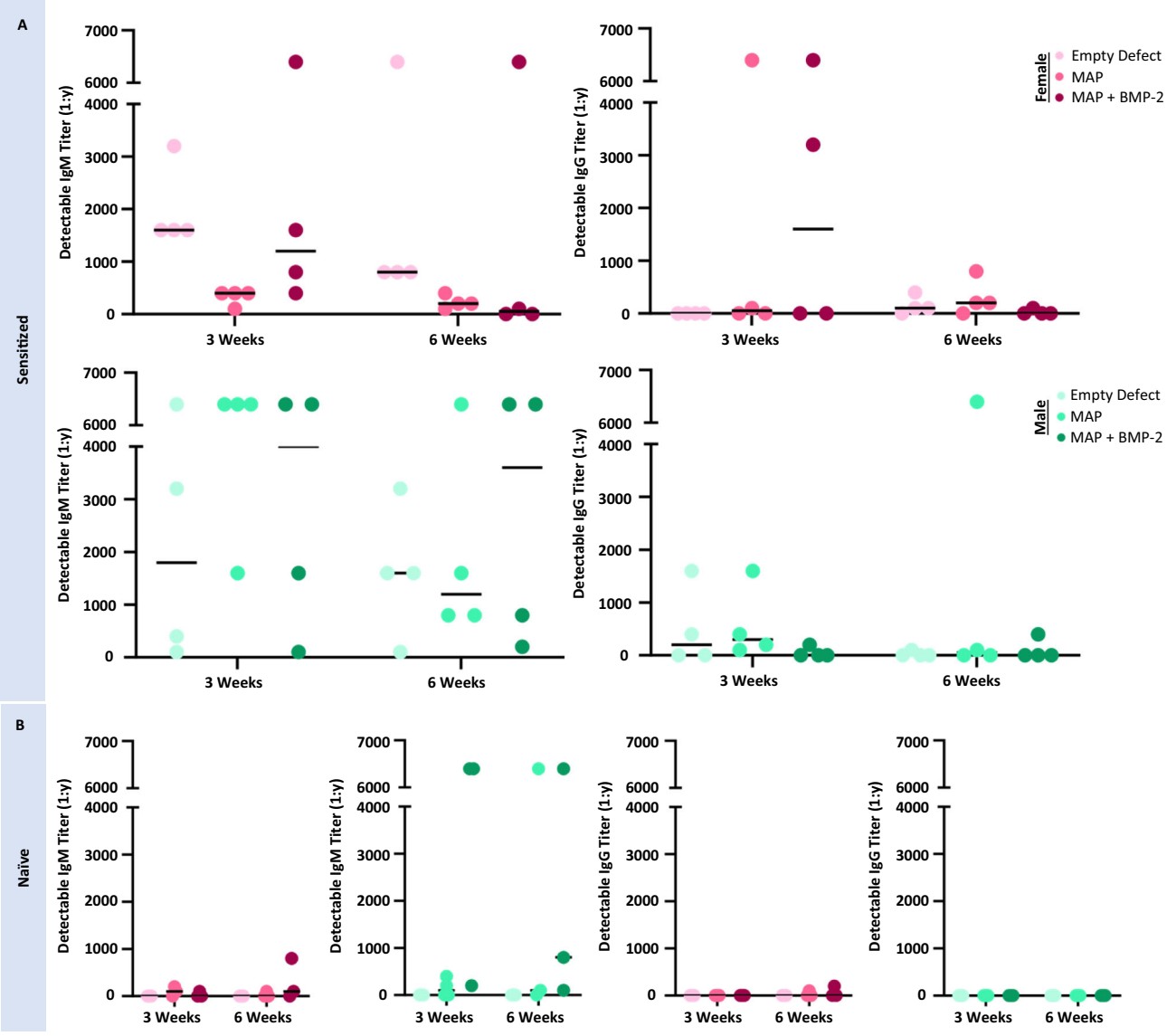

**Fig. 7 | Antibody formation persisted in PEG-sensitized mice and was detected in naïve mice after PEG hydrogel implantation. A** PEG sensitization contributed to IgM and IgG antibody formation post-implantation of treatment groups in sensitized mice. **B** Naïve mice also formed IgM antibodies post-implantation of PEG treatment groups. Dots represent individual specimens, and the black bars are the median of the sample group ($n = 4$ mice per sensitized treatment group; $n = 3$ mice per naïve treatment group).

images (Figs. 3 and 4 and Supplementary Figs. 4–7), new bone growth in the PEG sensitized mice treated with BMP-2 loaded PEG hydrogels protruded the boundaries of the original calvaria to a greater degree than what was observed in the PEG naïve mice with the same treatment. This observation raises concerns over ectopic bone growth, which is a known risk following BMP-2 use[69,70]. Additionally, inflammation has been associated with life-threatening complications in the off-label use of BMP-2 for cervical spinal fusion[71]. Cervical swelling in particular has been observed in several case studies where the surrounding soft tissue became inflamed and induced dysphagia[72,73]. The risks associated with using BMP-2 for bone regeneration have motivated research on novel biomaterial delivery systems, and PEG hydrogels have shown promising results. For example, Mariner et al. compared enzymatically degradable PEG hydrogels against absorbable collagen sponges, which are used clinically in INFUSE to deliver BMP-2, and found that the PEG hydrogels resulted in superior efficacy, even with a 10-fold lower dose of BMP-2[74]. However, our results suggest that patients with anti-PEG antibodies could be at risk for complications if PEG hydrogels are used as a vehicle for BMP-2 delivery.

Thus, if PEG-based delivery systems are translated from research to clinical use, our data suggests that it could be necessary to screen patients for anti-PEG antibodies.

In conclusion, this study is an important first step toward understanding the implications of anti-PEG antibody prevalence on PEG-based tissue engineering strategies. In considering the implications of our results, it is important to note that the anti-PEG titers in our mice were relatively high compared to what has been reported in humans. Clinical studies on anti-PEG antibody prevalence in the general population typically report anti-PEG antibody titer levels in the range of 1:80–1:100[18,75]. However, high anti-PEG titers comparable to the mice in our study have been reported, particularly in individuals treated with PEGylated pharmaceuticals. For example, patients treated with Krystexxa (PEGylated uricase) were reported to exhibit anti-PEGylated drug antibody titers greater than 1:2430[76]. Thus, the titers in our study are clinically relevant and represent patients who are at high risk for an anti-PEG reaction. Moreover, it was recently reported that the inclusion of PEG in lipid nanoparticle mRNA vaccines for SARS-CoV-2 stimulates anti-PEG antibody production and has a booster effect[25].

In a cohort of 130 individuals, the Pfizer-BioNTech and Moderna COVID-19 vaccines increased anti-PEG antibodies by a mean of 13.1- and 1.78-fold for IgG and 68.5- and 2.64-fold for IgM, respectively. Given the broad coverage of these vaccines, with over a billion doses administered worldwide, the prevalence of patients with high anti-PEG titers could increase in the future. Future studies should study anti-PEG titer effects and determine if there is a critical threshold. In addition, the gender effects in the immunological response to PEG identified in our study should be investigated further, and age effects should be investigated. Finally, the fundamental anti-PEG response to PEG hydrogels, which are the basis of multiple FDA-approved medical devices, and the impact on PEG hydrogels for other tissue engineering applications should also be investigated.

## Methods

### Study approval
Procedures were approved by Texas A&M University's Institutional Animal Care and Use Committee. Surgeries were conducted in accordance with animal use protocol 2018-0134.

### Materials
PEG−amide-tetranorbornene (PEGaNB) was synthesized via amidification of PEG−tetra-NH$_2$ (20 kDa; JenKem USA) with 5-norbornene−2-carboxylic acid (Sigma-Aldrich)[2]. Briefly, anhydrous dichloromethane (DCM; Sigma-Aldrich), 5 molar equivalents of N,N′-diisopropylcarbodiimide (DIC; Sigma-Aldrich), and 10 molar equivalents of 5-norbornene-2-carboxylic acid were mixed together in a reaction vessel for 45 min to form a norbornene anhydride. Meanwhile, PEG-tetra-NH$_2$ and 0.5 molar equivalents of 4-(dimethylamino)pyridine (DMAP; Sigma-Aldrich) were dissolved in anhydrous DCM in a round bottom flask and allowed to react for 5 min before 10 molar equivalents of pyridine was added and mixed for 15 min. Subsequently, the norbornene anhydride was filtered to remove urea salts, added to the reaction vessel with PEG-tetra-NH$_2$, and allowed to react overnight at room temperature. The reaction mixture was then precipitated in 10-fold volume excess chilled diethyl ether and filtered before being vacuum dried for 2 days, dialyzed against deionized water for an additional 2 days, and lyophilized. PEG-di-tetrazine was synthesized by functionalizing a PEG-diamine (3.4 kDa; Laysan Bio) with 5-(4-(1,2,4,5-tetrazin−3-yl)-benzylamino)-5-oxopentanoic acid (Tz-COOH; Sigma-Aldrich)[77]. Briefly, PEG-diamine was dissolved in 1-methyl-2-pyrrolidinone (NMP; Chem Impex) and mixed for 15 min with 2.5 molar equivalents of diisopropylethylamine (DIEA). Meanwhile, 3 molar equivalents of Tz-COOH were dissolved in NMP and activated for 5 min with 3 molar equivalents of O-(benzotriazol-1-yl)-N,N,N′,N′-tetramethyl-uronium hexafluorophosphate (HBTU; Chem Impex). The PEG-diamine and Tz-COOH solutions were then mixed together and allowed to react overnight at room temperature before precipitating the mixture in 10-fold volume excess of cold diethyl ether. The precipitate was vacuum dried for 2 days and subsequently dialyzed in deionized water for 2 days before lyophilization. Lithium acylphosphinate photoinitiator (LAP) was synthesized as described by Fairbanks et al.[78]. Briefly, dimethyl phenylphosphonite was mixed at room temperature under an argon blanket while a molar equivalent of 2,4,6-trimethylbenzoyl chloride was added dropwise. Following 18 h of the reaction proceeding, the mixture was quenched with a fourfold volume excess of lithium bromide. The solution was then heated for 10 min at 50 °C, cooled at room temperature for 4 h, filtration, and washed 3 times with 2-butanone. End-group functionalization of PEG − NB and PEG−di-tetrazine and LAP structure were confirmed by 1H NMR in DMSO-d$_6$ solvent. Cell adhesive peptide CGRGDS and matrix metalloproteinase (MMP) degradable crosslinker KCGPQGIAGQCK were synthesized with standard solid-phase Fmoc synthesis methods on Rink amide resin (Novabiochem)[44]. The resin was first swelled for 30 min

and underwent a deprotection step, 3 washes in NMP, and another deprotection step. Amino acids were subjected to a coupling reaction with an equimolar equivalent of HBTU, 3 NMP washes, capping reaction, 3 NMP washes, and a deprotection step. Deprotection steps were performed at 75 °C for 6 min using 20 wt% piperidine in NMP otherwise, except for aspartic acid (D), which was deprotected using 5 wt% piperazine in 0.1 M HOBt in NMP. Coupling reactions were generally performed at 75 °C for 8 min in 0.5 M in DIEA in NMP. The exceptions were cysteine (C), which was coupled at 50 °C for 6 min, and arginine (R), which was coupled at room temperature for 25 min and then 75 °C for 5 min. Capping steps were performed at 55 °C for 2 min in 10% acetic anhydride in NMP. After the final synthesis step was complete, the resin was washed in DCM. The peptides were then cleaved from the resin by a 2-h treatment with a trifluoroacetic acid (TFA):phenol:DI water:triisopropyl silane (90:5:2.5:2.5) cocktail and precipitated in chilled diethyl ether. The peptides were purified by high pressure liquid chromatography (ThermoFisher Dionex Ulti-Mate 3000) using water and acetonitrile with 0.1% trifluoroacetic acid as the mobile phase and C18 column as the stationary phase. The composition of the peptides was verified by matrix-assisted laser desorption−ionization time-of-flight mass spectroscopy (MALDI-TOF MS). Concentration of peptides was determined by measuring absorbance at 205 nm via NanoDrop OneC spectrophotometer (Thermo Scientific). All pre-polymer materials were confirmed to be endotoxin free (<0.5 EU/mL) by assessing endotoxin concentration with a kinetic LAL endotoxin assay (LONZA).

### PEG hydrogel microparticle synthesis
PEG microgels were fabricated via submerged electrospraying[79,80]. Briefly, PEGaNB, KCGPQGIAGQCK, CGRGDS and LAP were dissolved in endotoxin free and sterile PBS. Precursor solutions were mixed together at a 10 wt% working concentration of PEGaNB, 0.75:1 thiol-ene molar ratio, and 1 mM and 10 mM working concentration of CGRGDS and LAP, respectively. Pre-polymer solution was electrosprayed (12 mL/h, 4.0 kV, 22-gauge needle, 16 mm tip-to-ring distance) into light mineral oil (Fisher Scientific) with Span-80 (0.05 wt%) and photopolymerized with a 365 nm light at 60 mW/cm$^2$. Microgels were isolated and washed twice in sterile PBS, once in 70% ethanol, and twice in sterile PBS. Microgels were briefly vortexed and centrifuged at 3000 × $g$ for 15 min in between each washing step. Microgel diameter was measured using the line tool on ImageJ.

### PEG sensitization in C57BL/6 mice
Eight-week-old male and female C57BL/6 mice (Jackson Laboratory) were subcutaneously administered multi-PEGylated keyhole limpet hemocyanin (PEG-KLH, LifeDiagnostics) weekly for 4 weeks at a dosage of 2 mg/kg of body weight[38]. Naive mice were administered saline for the same duration of time and at the same dosage. One week after the final dosage, 10% of total blood volume was collected via submandibular bleed to confirm anti-PEG antibody presence. The mice were allowed to rest for 2 weeks following blood loss before performing calvarial defect surgery. For the duration of the PEG sensitization protocol, the mice were housed in groups of 4 according to sex and kept on 12 h light/12 h dark cycles at ambient temperature (20−22 °C) and humidity (40−60%).

### Calvarial defect surgery
Murine calvarial defects were performed in accordance with an animal use protocol (2018-0134) approved by the Texas A&M University IACUC. Mice were anesthetized under isoflurane inhalant in 100% oxygen and kept at 37 °C during surgery with a heating pad. Once anesthetized, mice were administered sustained release buprenorphine (dose = 1 mg/kg) via subcutaneous injection. Post-operation, mice were housed individually, routinely weighed, hydrated, and checked for surgical complications. After 5 days, the mice were housed

in groups of 4 according to sex. The environmental conditions were again 12 h light/12 h dark cycles at ambient temperature (20–22 °C) and humidity (40–60%).

To prepare the surgical site, Nair was used to remove hair followed by 3 washes each of chlorhexidine and 70% isopropyl alcohol, respectively. A unilateral 2.7 mm calvarial defect was performed 2–3 mm from the sagittal and coronal sutures using osteotomy burs (Strauss Diamond). The defects were filled with 6 μL of PEG microparticles and PEG-di-tetrazine (2 mM) with or without 500 ng of BMP-2, or left empty for the negative control groups. A total of 8 male mice and 8 female mice were prepared per treatment group per sensitization. The incision was closed with 2–3 nylon sutures and the mice were housed independently for 5 days followed by incision removal on day 7. At 21 and 42 days post operation, the mice were euthanized via $CO_2$ euthanasia and terminal cardiac bleed. A total of 4 male mice and 4 female mice per treatment group per sensitization were euthanized at each time point. Plasma was obtained from the mice during the euthanasia time point.

### Anti-PEG antibody ELISA

Detectable titers of anti-PEG IgG and IgM isotypes were determined via ELISA with slight modifications to methods described in previous studies[38]. First, 100 μL of 10 μg/mL 20 kDa multi-PEGylated bovine serum albumin (PEG-BSA, Life Diagnostics) was added to each well in a high binding 96-well plate and incubated at 4 °C overnight. After incubation, a wash step was performed (5 washes with 300 μL PBS). Next, 100 μL of block solution (1% bovine serum albumin (BSA) in PBS) was added to each well and incubated for 1 h at room temperature. After a wash step, 100 μL of mouse plasma diluted in diluent (1% BSA in PBS) at serial dilutions starting from 1:100 to 1:64,000 was added to each well and incubated at 37 °C for 1 h followed by a wash step. 100 μL of IgG (Invitrogen, catalog # A16084, lot # 93-40-080422, polyclonal, 1:150), or IgM (Invitrogen, catalog # 62-6820, lot # YB372990, polyclonal, 1:50) HRP conjugated secondary antibody diluted in diluent was added to each well and incubated for 1 h at room temperature. After a wash step, 100 μL of TMB was added to each well and allowed to develop for 20 min at room temperature in the dark. In all, 100 μL of stop solution was added and absorbance was measured at 450 nm and corrected at 570 nm. The highest detectable titer with an absorbance value greater than the average absorbance + 2 standard deviations of the plasma samples from naive mice pre-implantation was determined to be the detectable titer.

### Characterization of cytokine production

Cytokines in circulation were determined using a cytokine/chemokine magnetic bead panel (MILLIPLEX). Sample preparation was performed according to manufacturer protocol. Briefly, wells in a 96-well plate were incubated with 200 μL of wash buffer for 10 min at room temperature. Wash buffer was decanted, and 25 μL of standards, controls, and background were added to the appropriate wells. 25 μL of assay buffer was added to wells that would contain samples. 25 μL of serum matrix were added to wells containing standards, control, and background. 25 μL of sample (1:1 serum sample to assay buffer) were added to the other wells, and 25 μL of mixed beads were added to all wells. The plate was sealed and incubated in the dark with agitation overnight at 4 °C. Well contents were washed twice before 25 μL of detection antibodies were added to each well. Contents were incubated in the dark with agitation for 1 h at room temperature. 25 μL of streptavidin–phycoerythrin was added to each well and allowed to incubate for 30 min under the aforementioned conditions, and the well contents were washed twice before 150 μL of Drive Fluid PLUS was added to each well. Well contents were analyzed using the Luminex software.

### Tissue sample preparation

Following euthanasia, the area above the calvaria was removed of hair and cleaned using 70% ethanol. A single, longitudinal incision was made along the suture line of the mouse skull. The skin and fascia were separated from the skull, and an incision was made around the defect site and a portion of the contralateral side using a rotary blade (Strauss Diamond). A rectangular portion of the calvarium, containing the defect side and an equal size on the contralateral side, was removed, and placed in 10% neutral buffered formalin (NBF, Sigma-Aldrich) for 48 h. This procedure was performed for each animal subject. Calvarial samples were then gently washed in 1× PBS before being placed in Carson's fixative (10% NBF, 1.86 wt% $Na_2HPO_4$, 0.42 wt% NaOH) and stored at 4 °C until bone quantification.

### Microcomputed tomography (μCT) analysis

Following fixation, bone formation was quantified using μCT (Bruker Skyscan 1275, software version 1.0.15). Briefly, samples were washed in PBS and subsequently wrapped in Parafilm before being scanned. Calvarial samples were scanned at a voltage of 30 kV, current of 200 μA, and pixel resolution of 18 μm. Scans were performed at 0.5° increments over a total of 360°, and each scanned increment was based on the average of 3 frames per 0.5° rotation. Scans were then used to reconstruct a 3D image. Parameters like smoothing, ring artifact reduction, and beam hardening were maintained at 2, 5, and 41%, respectively, for all samples. Misalignment compensation was adjusted to minimize artifacts among each sample and ranged from −3.5 to 1.5. Samples were normalized to a phantom sample's dynamic range (max attenuation coefficient) from 0 to 0.080838. Axial images were then reconstructed and oriented for quantification using NRecon software (version 1.7.4.6). Using CTAn software (version 1.16.4.1), image compilations of each sample were subsequently adjusted to a minimum and maximum attenuation calibration of 0.00312 and 0.09452, respectively. One hundred and one z-slices were used to quantify bone volume and bone surface area per sample.

### Sample decalcification

Following bone tissue quantification, samples were gently washed in PBS before being placed in 0.5 M EDTA (pH - 7.2) and stored at 4 °C. Samples were placed in fresh EDTA solution every 2–3 days.

### Histology and immunohistochemistry

Prior to dehydration and paraffin embedding, the area around the defect was marked with a tissue marking dye (Triangle Biomedical Sciences) to aid in identifying the defect area during microtome sectioning. The marking solution was allowed to dry before starting the dehydration process. The tissues were submerged for 1 h incubations in the following order: 95% ethanol, 95% ethanol, 80% ethanol, 60% ethanol, 3 incubations in xylene, 1:1 xylene/paraffin, and paraffin. After dehydration, the tissues were embedded in paraffin using tissue molds, followed by sectioning in 6 μm-thick sections using a Shandon Finesse 325 microtome (Thermo Scientific). The tissue sections were then mounted on silane-treated glass slides.

For hematoxylin and eosin (H&E) staining, the tissue sections were treated with a 3 min and 2 min incubation in xylene, followed by 30 s incubations in 100%, 95%, 80%, 60%, and 30% ethanol. After a 30 s tap water rinse, the sections were incubated in hematoxylin (Biocare Medical) for 2 min followed by a tap water rinse. The sections were dipped 3 times in 1% acetic acid, washed 3 times in tap water, once in PBS for 1 min, and followed with 3 washes in tap water. After 1 min incubations in 85% and 95% ethanol the sections were incubated in eosin-y (StatLab) for 30 s. Excess eosin-y was washed away in 1 min washes of 95% and 100% ethanol, and the slides were cleared with 2, 1 min incubations with xylene.

For Masson's Trichrome staining, the STATLAB Masson's Trichrome kit was used. Staining was performed according to the manufacturer's protocol. Briefly, sections were treated with xylene twice for 5 min, followed by three 1 min washes in 100% EtOH. Sections were subsequently washed in tap water for 1 min and incubated overnight at room temperature in Bouin's fluid. Following this overnight incubation step, the sections were rinsed with tap water for 3 min and incubated in a working solution of Weigert's hematoxylin (1 part Weigert's hematoxylin A, 1 part Weigert's hematoxylin B) for 5 min Following a 2 min wash step, the sections were incubated in biebrich scarlet-acid fuchsin for 15 min, followed by a 1 min wash in tap water and subsequent incubation in phosphomolybdic/phosphotungstic acid for 15 min. The sections were next incubated in aniline blue stain for 10 min, washed for 1 min in tap water, incubated in 1% acetic acid for 5 min, and incubated in 100% EtOH for three rounds of 1 min. Finally, the sections were cleared with xylene for 1 min.

For immunohistochemistry, tissue sections were treated with Proteinase K (Abcam) for 10 min and washed 3 times with TBS with 0.025% Triton X (TBST) for 2 min Sections were then blocked with 5% goat serum in TBST, and they were incubated in anti-CD3 antibody (Abcam, catalog # ab5690, lot # YB372990, polyclonal, 1:300) or anti-CD68 antibody (Invitrogen, catalog # PA5−78996, lot # YB372990, polyclonal, 1:800) in 1% BSA in TBST overnight at 4 °C. Sectioned were washed 3 times in TBST for 2 min before incubation in 3% $H_2O_2$ for 15 min and followed with 3 washed in TBST. HRP conjugated Goat Anti-Rabbit IgG H&L secondary antibody (Abcam, catalog # ab205718, lot # YB372990, polyclonal, 1:1000) in 5% BSA in TBST was applied to slides and allowed to incubate for 1 h at room temperature before 3 washes in TBST. Sections were then incubated in DAB for 7 min, washed in tap water for 5 min, and incubated with 1% methyl green counterstain (Possible Missions Inc.) for 3 min. Tissue sections were dehydrated in increased concentrations of ethanol and cleared in xylene for 1 min. For quantification of CD3+ and CD68+ positive cells, 4 tissue sections were imaged at ×10 (Motic Easy Scan) from each animal and number of CD3+ and CD68+ cells were counted and reported as number of positive cells per mm$^2$.

For all stains, a glass coverslip was mounted to the surface using 2 drops of Permount mounting medium (Fisher Chemical) and allowed to cure overnight. Tissue sections were imaged at ×4 and ×10 utilizing a Biotek Lionheart automated microscope (Agilent).

### Collagen quantification

Paraffin-embedded tissue samples were sectioned at a 6μm thickness and collected in 50μm increments across the defect site. Collected tissue sections were mounted on silane treated glass slides and subjected to Masson's trichrome staining. Collagen at the defect site for each section was imaged (Motic Easy Scan version 1.0.5.134) and quantified via ImageJ, and the percent area of collagen deposition from each tissue section was averaged for each animal sample.

### Statistical analysis

Statistical analysis of the quantitative data was performed using GraphPad Prism (version 10.1.2). Statistical significance of quantitative μCT data, collagen deposition data, and quantitative IHC data were determined by two-way ANOVA with Tukey's post hoc tests.

### Statistics and reproducibility

The PEG sensitization protocol using subcutaneous injections of PEG-KLH has been replicated eight times independently with similar results. The calvarial defect study was performed once using independent biological quadruplicates but was not otherwise replicated. ELISAs were measured and analyzed as technical duplicates and biological triplicates and quadruplicates for naive and sensitized groups, respectively. Cytokine production was measured and analyzed as technical duplicates and biological quadruplicates.

### Reporting summary

Further information on research design is available in the Nature Portfolio Reporting Summary linked to this article.

## Data availability

All data needed to evaluate the conclusions in the paper are provided in the Supplementary Information. Source data are provided with this paper and are available online in the Figshare repository (https://doi.org/10.6084/m9.figshare.24769155).

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

## Acknowledgements

Research reported in this publication was supported by the National Institute of Arthritis and Musculoskeletal and Skin Diseases (R21AR074635 to D.L.A.), the National Institute of General Medical Sciences (R01GM147821 to D.L.A.), and the National Eye Institute (R01EY029350 to R.H.L.) of the National Institutes of Health. A.H.I. was supported by K12 GM111726 and the National Science Foundation Graduate Research Fellowship Program. The content is solely the responsibility of the authors and does not necessarily represent the official views of the National Institutes of Health or National Science Foundation. The authors thank Dr. Abhishek Jain of the Texas A&M University Department of Biomedical Engineering for the use of his lab's Luminex MAGPIX analyzer. Schematics were created with Biorender.com.

## Author contributions

Conceptualization: A.H.I. and D.L.A.; investigation: A.H.I., S.Y.R.P., E.R., M.E., V.R., T.B. and C.P.; methodology: A.H.I., S.Y.R.P., R.H.L., M.F.C., C.G. and D.L.A.; formal analysis: A.H.I. and S.Y.R.P.; visualization: A.H.I. and S.Y.R.P.; writing—original draft: A.H.I. and S.Y.R.P.; writing—review and editing: A.H.I., S.Y.R.P., K.L., T.G., R.H.L., M.F.C., C.G., D.L.A.; resources: R.H.L., T.G., C.G., D.L.A.; funding acquisition: A.H.I., R.H.L. and D.L.A.; supervision: D.L.A.; project administration: D.L.A.

## Competing interests

The authors declare no competing interests.
