## [Peer Review File · Nature Communications]

Reviewers' Comments:

Reviewer #1:

Remarks to the Author:

Overall Comment/Reflection:

Isaac et al introduce a critically necessary question: can we still use PEG-based regenerative materials in an organism with anti-PEG antibodies? The first paper of its kind, this work introduces the need to consider PEG-based tissue engineering products in the context of growing anti-PEG antibodies in the general population. The findings from the paper are somewhat lackluster, however, as the mechanism and robust characterization of the results are limited. Enthusiasm for the scientific question is higher than enthusiasm for the results shared. In particular, the histological results are difficult to interpret, the influence of cellular invasion on bone defect repair is not addressed, and the lack of immunological characterization weakens results from the proposed manuscript.

Large-Scale Concerns/Comments:

- Introduction:

- o Page 4 line 8: general population of what? Please clarify what population this is referring to.
- o The phrase "PEG is ubiquitous in our environment" is vague. An added citation could strengthen this claim detailing the products and exposure of populations to PEG.
- o The introduction of PEG-KLH is not presented well or clearly. An introductory sentence to the purpose and relevance of this is needed.
- o A reason for choosing MAPs vs other kinds of PEG depots is needed for context and relevance to the MAP PEG formulation.
- o The last sentence... "changes in bone growth and inflammation..." no studies related to inflammation were presented in this work. This phrase should be removed.

- The element of age of biological sex:

- o How is the element of age playing a role in the immunological response of biological sex differences? This is not addressed and the rationale for the age of the mice is not provided.
- A study to prove that inflammatory cytokine production is amplified due to PEG sensitization in their models is pertinent

- In general, all figure captions can be more descriptive.

- Figure 1:

- o One thing that could be clearer is in figure 1, when did the tissue processing and data analysis happen? Was that right after the 6-week time point? Please edits for clarity.
- o The reason behind choosing a calvarial defect is unclear. Please provide a rationale for the bone defect.

- Figs 3,4: it's necessary to introduce quantified graphs of histological data. What is the quantity of collagen in Figure 4? As it is right now, it is difficult for one to assess exactly what is good or bad. Do we want more porosity or less, is woven structure bad?

- o It is not clear what healthy/normal bone in this region should look like, either. The results from these figures are impossible to articulate.

- o Labeling of the defect site is needed.

- Interrogation of cells and cytokines in the bone defect:

- o The presence of immune cells could clearly be investigated in each group via histology. What populations of immune cells are present at each time point? A simple CD45+ staining could help address underlying differences across each group and biological sex.

- Statistical analysis: the number of mice per group must be reported for each analysis and in each figure. How many mice are used in each assessment?

- o Additionally, a rationale for the age of the mice is needed.

Minor Comments/Edits:

- Results and Discussion pg 5, line 1- delivery from PEG-based MAP gels was found to be significantly impacted in PEG-sensitized mice... : it should be mentioned whether it is positively or negatively impacted?
- Fig S2: what could be the reasoning behind male-sensitized mice showing such different levels of antibody titer?

- Page 7, line 6- the group mentions morphological differences between the calvarial tissue to the contralateral side. What could be the reasons to these differences observed?
- Page 8, lines 7-0: how are mesenchymal cells related to MMP expression?
- Line 6- no fig 6 in the paper. Remove from text
- Page 13, line 2-4: how can it be differentiated that the immune response or inflammation is due to MAP+BMP2 vs sutures on the site of incision?
- Page 15, Statistical analysis: were determined by two-way ANOVA with post hoc comparisons... please include which post-hoc comparisons? Bonferroni or Tukey or something else
- Fig 2: 6 wks, why does a sensitized male have more bone volume? This is postulated but not interrogated in the manuscript.
- Fig 3: how many mice were used? How was the data quantified?

Reviewer #2:

Remarks to the Author:

In their manuscript "Impact of PEG Sensitization on the Efficacy of PEG Hydrogel-Mediated Delivery of Bone Morphogenetic Protein 2", the authors report that antibodies against polyethylene glycol (PEG) can accelerate regeneration of bone defects in mice treated with a PEG hydrogel containing BMP-2 but that the resulting bone structure was abnormal. These observations are new and interesting.

While the observations are intriguing, the study currently lacks important mechanistic details. To be considered for publication, I think that some additional studies on how the antibodies are accelerating bone growth are warranted.

Major points

1. The authors posit that the anti-PEG antibodies increase hydrogel degradation and accelerate release of BMP-2 from the hydrogel but this is not shown. I think this key hypothesis should be tested experimentally, at least by showing how much BMP-2 remains in the hydrogel. The results at 6 weeks (Fig 2, middle panel) show that in male mice, anti-PEG antibodies in untreated mice or mice with empty biogel caused a similar percentage increase in bone volume as the mice with loaded biogel. How do these results fit into this hypothesis?
2. The authors suspect that increased MMP concentrations may have caused the more rapid degradation of the MMP-degradable polypeptide cross-linker. Is there a way to compare MMP amount or activity in the biogel implant site to support this hypothesis? An assay using the MMP-sensitive polypeptide present in the biogel seems feasible. Do the authors think that anti-PEG antibodies would affect the degradation of biogels without MMP cross-linkers?
3. The class of the anti-PEG antibodies should be reported (IgG versus IgM). This is easy to do and potentially important in the mechanism of action. Was there any difference in anti-PEG titers between male and female mice? This may explain some of the sex differences.
4. The authors used two-way ANOVA for pairwise comparisons and removed "outliers" from sample sets if values were above 1.5x the interquartile range (IQR) of the 3rd quartile or below 1.5xIQR of the 1st quartile. This seems inappropriate for such small sample sizes (n=4). It tends to make things appear artificially significant. I think that performing the student t test on the intact data set is more appropriate in this case when you are comparing mean values between two groups at a time.

Minor points

1. Figure legends could use more details so the reader can understand the results without having to refer to the text.

2. It is a little difficult to tell which groups are being compared in the statistical comparisons in Figure 2. Maybe the authors can put small downward marks on the black lines to make the comparison easier to see.

3. The authors rightly point out the seminal PEG study by Richter et al, "Richter et al. reported that injecting PEGylated ovalbumin in mice caused an increase in anti-PEG IgM antibodies and a rapid clearance of a second dose of the same PEGylated protein.

Actually, most of the experiments in that paper used rabbit. I don't think they measured clearance, but rather observed symptoms of anaphylaxis in guinea pigs sensitized with rabbit anti-PEG serum. As far as I know, Richter also didn't distinguish between IgM and IgG. The first report of a pegylated protein ABC was probably: Cheng et al. Accelerated clearance of polyethylene glycol-modified proteins by anti-polyethylene glycol IgM. *Bioconj. Chem.* 10: 520-528, 1999.

4. More details of the assay used to measure anti-PEG antibody titer would be useful.

Reviewer #3:

Remarks to the Author:

The objective of this work is to study how pre-existing PEG antibodies could change the outcomes of bone regeneration utilizing PEG-based hydrogels. To achieve the goal, the authors immunized C57BL/6 mice against PEG and performed a bone tissue engineering study in which calvarial defects are treated with bone morphogenetic protein-2 loaded PEG microporous annealed particle hydrogels to promote bone regeneration.

1. It is important that the authors use another set of non-PEG hydrogels loaded with bone morphogenetic protein-2. Otherwise, the data shown in Figure 2B are not convincing. There are many reasons that can cause the difference in data in Figure 2B

2. Although the authors provide a lot of reasons to explain the difference, there are no direct correlations or evidences between anti-PEG antibodies generated and bone regeneration. At its form, it only reports what is observed with speculations. Without additional controlled experiments as described above, it is hard to make the connection.

3. Furthermore, the difference between male and female mice could be simply due to the difference from various possibilities such as a different level of anti-PEG antibodies generated, the different drug delivery profiles from these microgels and more.

4. Writing can be improved by paying attention to the detail. For example, there is no Figure 6 as cited in the text and Figure 2B caption should be better described.

REVIEWER COMMENTS

Reviewer #1 (Remarks to the Author):

Overall Comment/Reflection:

Isaac et al introduce a critically necessary question: can we still use PEG-based regenerative materials in an organism with anti-PEG antibodies? The first paper of its kind, this work introduces the need to consider PEG-based tissue engineering products in the context of growing anti-PEG antibodies in the general population. The findings from the paper are somewhat lackluster, however, as the mechanism and robust characterization of the results are limited. Enthusiasm for the scientific question is higher than enthusiasm for the results shared. In particular, the histological results are difficult to interpret, the influence of cellular invasion on bone defect repair is not addressed, and the lack of immunological characterization weakens results from the proposed manuscript.

Large-Scale Concerns/Comments:

- Introduction:
 - Page 4 line 8: general population of what? Please clarify what population this is referring to.
Thank you for this question. We have revised this statement to clarify this study evaluated anti-PEG antibody prevalence “in a sample size of 377 healthy individuals”
 - The phrase “PEG is ubiquitous in our environment” is vague. An added citation could strengthen this claim detailing the products and exposure of populations to PEG.
Thank you for this comment and suggestion. We have added the following citation to support our statement:

Jang, H.-J., Shin, C. Y. & Kim, K.-B. Safety evaluation of polyethylene glycol (PEG) compounds for cosmetic use. Toxicological research 31, 105-136 (2015).
 - The introduction of PEG-KLH is not presented well or clearly. An introductory sentence to the purpose and relevance of this is needed.
Thank you for this suggestion. Background information and the justification for utilizing PEG conjugated KLH for PEG sensitization are now included in the Introduction.
 - A reason for choosing MAPs vs other kinds of PEG depots is needed for context and relevance to the MAP PEG formulation.
Thank you for this suggestion. We chose MAP hydrogels because they can easily be formed within tissue defects and their network of interconnected pores has been shown to result in superior tissue regeneration when compared to

conventional, nanoporous PEG hydrogels. This justification has been included in the Introduction.

- The last sentence... “changes in bone growth and inflammation...” no studies related to inflammation were presented in this work. This phrase should be removed.

Thank you for this comment. Since we have now included data on immune cell recruitment, we have revised this statement to “changes in bone growth, the histological response, and immune cell recruitment.”

- The element of age of biological sex:
 - How is the element of age playing a role in the immunological response of biological sex differences? This is not addressed and the rationale for the age of the mice is not provided.

This is an interesting question. We chose to use young adult mice to ensure a robust anti-PEG response. Additionally, we designed our study so that the mice would be approximately the same age as in our other bone regeneration work when we performed our calvarial defect surgery. We did not investigate the effects of age, but we now acknowledge this is an important direction for future work in the Conclusions.

- A study to prove that inflammatory cytokine production is amplified due to PEG sensitization in their models is pertinent

Thank you for this suggestion. In response, we tested mouse plasma samples that were collected at 3 and 6 weeks for inflammatory cytokines via a Luminex assay. We specifically tested samples from the naïve and sensitized male groups treated with MAP+BMP-2. The results showed that there were no systemic changes in inflammatory cytokine production, regardless of PEG sensitization. We believe this is a noteworthy finding and have added it to the manuscript. However, it obviously does not preclude the possibility that there were changes in cytokine production locally at the defect site. Unfortunately, we are unable to test for this at this time.

- In general, all figure captions can be more descriptive.

Thank you for the suggestion. We have updated the figure captions to be more descriptive.

- Figure 1:

- One thing that could be clearer is in figure 1, when did the tissue processing and data analysis happen? Was that right after the 6-week time point? Please edits for clarity.

Thank you for this question. Tissue processing and data analysis were performed immediately after tissue sample collection, which was at 3 and 6 weeks post-implantation. This information has been added to the **Figure 1** caption.

- The reason behind choosing a calvarial defect is unclear. Please provide a rationale for the bone defect.

Thank you for this question. We have revised the caption to clarify that the calvarial defect model is widely used to evaluate the efficacy of biomaterials and therapeutics for bone regeneration. This is supported by the references below. Additionally, this particular defect model allows for the assessment of bone regeneration without interference from additional variables such as mechanical loading, which would be present in a long-bone defect model (e.g., femur).

Schmitz, John P., and Jeffrey O. Hollinger. "The critical size defect as an experimental model for craniomandibulofacial nonunions." *Clinical Orthopaedics and Related Research* (1976-2007) 205 (1986): 299-308.

Gomes, P. S., and M. H. Fernandes. "Rodent models in bone-related research: the relevance of calvarial defects in the assessment of bone regeneration strategies." *Laboratory Animals* 45.1 (2011): 14-24.

- Figs 3,4: it's necessary to introduce quantified graphs of histological data. What is the quantity of collagen in Figure 4? As it is right now, it is difficult for one to assess exactly what is good or bad. Do we want more porosity or less, is woven structure bad?

Thank you for this comment. Collagen deposition was quantified from the Masson's Trichrome staining. This data is presented in **Figure 5** and suggests significant differences between treatment groups, particularly at the 3-week time point. Regarding the structure, the *in vivo* model used in the study relies on intramembranous ossification to form bone, and the early stage of this specific bone formation process generates a woven bone matrix, as described in the reference below. The mature bone formed through this type of ossification is typically flat, similar to the morphology of the contralateral side represented in Figures 3 and 4.

Moreira, Carolina A., David W. Dempster, and Roland Baron. "Anatomy and ultrastructure of bone—histogenesis, growth and remodeling." (2015).

- It is not clear what healthy/normal bone in this region should look like, either. The results from these figures are impossible to articulate.
As noted above, healthy bone can be seen on the contralateral side. This is visible in the low magnification images in **Figures 3 and 4** panels, which show the entire tissue sections.
- Labeling of the defect site is needed.
The defect sites for each representative tissue sample in **Figures 3 and 4** are indicated via solid-line boxes. This is stated in the figure captions.

- Interrogation of cells and cytokines in the bone defect:
 - The presence of immune cells could clearly be investigated in each group via histology. What populations of immune cells are present at each time point? A simple CD45+ staining could help address underlying differences across each group and biological sex.
Thank you for this suggestion. To evaluate the presence of immune cells, we chose to perform immunohistochemistry for CD3 and CD68 positive cells at both time points investigated in the study. IHC images and quantitative data are represented in **Figure 6** and discussed in the manuscript.

- Statistical analysis: the number of mice per group must be reported for each analysis and in each figure. How many mice are used in each assessment?
Each treatment group included 4 mice. The number of mice per group is stated in the methods and the figure captions.
 - Additionally, a rationale for the age of the mice is needed.
As noted in a prior comment, we chose to use young adult mice to ensure a robust anti-PEG response. This is clarified in the Methods.

Minor Comments/Edits:

- Results and Discussion pg 5, line 1- delivery from PEG-based MAP gels was found to be significantly impacted in PEG-sensitized mice... : it should be mentioned whether it is positively or negatively impacted?
Thank you for noting this. We have substantially revised this section of our manuscript and are now more specific about how bone formation was impacted.

- Fig S2: what could be the reasoning behind male-sensitized mice showing such different levels of antibody titer?
This is an interesting question. All mice, regardless of gender were administered the same dosage of PEG-KLH. We are unsure of the reason, but our data clearly indicate biological variability between animals and sex-dependent differences. Similar variability and sex differences have been reported in the literature for humans.

- Page 7, line 6- the group mentions morphological differences between the calvarial tissue to the contralateral side. What could be the reasons to these differences observed?
Thank you for this question. A critical-sized defect was not created in the contralateral side, so there was no injury and MAP hydrogels were not implanted.

- Page 8, lines 7-0: how are mesenchymal cells related to MMP expression?
As noted earlier, we have substantially revised this section of our manuscript. The statement in question has been removed.
- Line 6- no fig 6 in the paper. Remove from text
Thank you for noting this. The text has been removed.
- Page 13, line 2-4: how can it be differentiated that the immune response or inflammation is due to MAP+BMP2 vs sutures on the site of incision?
Thank you for this question. All mice had their incisions closed with sutures. If there had been an immune response or inflammation due to the suture, the data would have reflected this in all of the treatment groups, including the empty defect. Additionally, no infection or use of antibiotics in response to infection or inflammation around the sutures was documented for any mice during post-op checks.
- Page 15, Statistical analysis: were determined by two-way ANOVA with post hoc comparisons... please include which post-hoc comparisons? Bonferroni or Tukey or something else.
Thank you for this question. Tukey post-hoc comparisons were performed. This has been clarified in the methods section.
- Fig 2: 6 wks, why does a sensitized male have more bone volume? This is postulated but not interrogated in the manuscript.
Thank you for this question. As mentioned earlier, we added IHC quantification of CD3 and CD68 positive cells at the defect site to detect cells relevant to the immune response and regeneration. This data is presented in **Figure 6**. Significant differences were noted, and the different T cell response noted in males may have contributed to the differences in bone volume. This data is discussed in the text.
- Fig 3: how many mice were used? How was the data quantified?
A total of 4 mice per experimental group were stained with H&E. No quantification was performed on the H&E images, but collagen deposition has been quantified from Masson's Trichome images. The quantification method is described in the Methods section, and the data is presented in **Figure 5**.

Reviewer #2 (Remarks to the Author):

In their manuscript "Impact of PEG Sensitization on the Efficacy of PEG Hydrogel-Mediated Delivery of Bone Morphogenetic Protein 2", the authors report that antibodies against polyethylene glycol (PEG) can accelerate regeneration of bone defects in mice treated with a PEG hydrogel containing BMP-2 but that the resulting bone structure was abnormal. These observations are new and interesting.

While the observations are intriguing, the study currently lacks important mechanistic details. To be considered for publication, I think that some additional studies on how the antibodies are accelerating bone growth are warranted.

Major points

1. The authors posit that the anti-PEG antibodies increase hydrogel degradation and accelerate release of BMP-2 from the hydrogel but this is not shown. I think this key hypothesis should be tested experimentally, at least by showing how much BMP-2 remains in the hydrogel. The results at 6 weeks (Fig 2, middle panel) show that in male mice, anti-PEG antibodies in untreated mice or mice with empty biogel caused a similar percentage increase in bone volume as the mice with loaded biogel. How do these results fit into this hypothesis?

Thank you for this comment and suggestion. While we believe that the degradation of the hydrogels may have been impacted by PEG sensitization, quantifying the amount of BMP-2 remaining in the hydrogels is non-trivial and cannot be done with the tissue samples we have available. While it could potentially be done by labeling the BMP-2 with a near-infrared excitable fluorophore and performing a live-animal imaging study (DOI: 10.1016/j.actbio.2017.06.028), this is not within our capabilities at this time. Regarding the bone volume measurements in the male mice at 6 weeks, it is interesting that the percent differences between the sensitized and control groups are similar for all three treatments. Although these differences were not statistically significant, it is possible that differences immune cell recruitment in the PEG sensitized group could be having an impact.

2. The authors suspect that increased MMP concentrations may have caused the more rapid degradation of the MMP-degradable polypeptide cross-linker. Is there a way to compare MMP amount or activity in the biogel implant site to support this hypothesis? An assay using the MMP-sensitive polypeptide present in the biogel seems feasible. Do the authors think that anti-PEG antibodies would affect the degradation of biogels without MMP cross-linkers?

Thank you for this comment. Similar to quantifying BMP-2 release/retention, accurately quantifying hydrogel degradation in vivo is non-trivial. It could potentially be done by using fluorescence or magnetic resonance imaging (DOI: 10.1021/acsbomaterials.0c01547), but this would require synthetic modifications as well as advanced instrumentation and is beyond our current capabilities. It is not possible to analyze MMP activity in our current tissue samples, and in vitro experiments would not recapitulate immune and bone cell recruitment in vivo. With that said, we did examine the recruitment of CD68+ cells (macrophages and osteoclasts), which are

a source of MMPs in vivo. Higher numbers of CD68+ cells were observed in sensitized mice treated with MAP + BMP-2 at 3 weeks, and sensitized females had a significantly higher number of CD68+ cells at 6 weeks compared to their naïve counterparts. This data is presented in **Figure 6**, and the Results and Discussion section has been revised accordingly.

We would not expect gels without MMP-degradable crosslinkers to be affected. We specifically chose a 4-arm PEG-NB with an amide linkage instead of an ester to prevent MAP hydrogel degradation by hydrolysis, which would normally be an alternative mechanism of degradation in vivo. While oxidative degradation of the PEG ether bonds is possible, this occurs on a longer timescale. Due to our materials, selection we do believe the only mechanism of degradation would be by MMPs.

3. The class of the anti-PEG antibodies should be reported (IgG versus IgM). This is easy to do and potentially important in the mechanism of action. Was there any difference in anti-PEG titers between male and female mice? This may explain some of the sex differences.

Thank you for this suggestion. We agree. IgG and IgM titers of the mice were determined pre- and post-implantation for all treatment types for males and females via ELISAs. Pre-implantation data is represented in **Supplemental Figure 2**, and post-implantation data is represented in **Figure 7**. Pre-implantation data demonstrated higher IgM titers in males than their female counterparts comparable IgG titers. These trends persisted among the sensitized mice at 3 and 6 weeks post-implantation. Naïve mice also demonstrated increased IgM titers post-implantation, particularly in the MAP + BMP-2 group.

4. The authors used two-way ANOVA for pairwise comparisons and removed “outliers” from sample sets if values were above 1.5x the interquartile range (IQR) of the 3rd quartile or below 1.5xIQR of the 1st quartile. This seems inappropriate for such small sample sizes (n=4). It tends to make things appear artificially significant. I think that performing the student t test on the intact data set is more appropriate in this case when you are comparing mean values between two groups at a time.

Thank you for this comment and suggestion. While we do not believe that exclusion of statistical outliers is unreasonable, we have decided to include the outlier that was previously excluded using the 1.5x IQR method in **Figure 2** and comment on this data point in the text. Regarding the statistical analysis performed on our data sets, we disagree that a t-test would be appropriate and believe that ANOVA followed by post hoc Tukey comparisons is more appropriate.

Minor points

1. Figure legends could use more details so the reader can understand the results without having to refer to the text.

Thank you for this suggestion. Details have been added to both the figure legends and captions for all figures.

2. It is a little difficult to tell which groups are being compared in the statistical comparisons in Figure 2. Maybe the authors can put small downward marks on the black lines to make the comparison easier to see.

Thank you for the suggestion. Downward marks have been added to clarify the statistical comparisons.

3. The authors rightly point out the seminal PEG study by Richter et al, "Richter et al. reported that injecting PEGylated ovalbumin in mice caused an increase in anti-PEG IgM antibodies and a rapid clearance of a second dose of the same PEGylated protein.

Actually, most of the experiments in that paper used rabbit. I don't think they measured clearance, but rather observed symptoms of anaphylaxis in guinea pigs sensitized with rabbit anti-PEG serum. As far as I know, Richter also didn't distinguish between IgM and IgG. The first report of a pegylated protein ABC was probably: Cheng et al. Accelerated clearance of polyethylene glycol-modified proteins by anti-polyethylene glycol IgM. *Bioconj. Chem.* 10: 520-528, 1999.

Thank you for these points. The correct citation has been added to the text.

4. More details of the assay used to measure anti-PEG antibody titer would be useful.

This anti-PEG antibody ELISA protocol was modified from Li, B. et al. (cited as reference 38) and is now included in detail in the methods section. We have also determined the detectable titers in all sensitized mice and present this data in **Supplemental Figure 2** and **Figure 7**.

Reviewer #3 (Remarks to the Author):

The objective of this work is to study how pre-existing PEG antibodies could change the outcomes of bone regeneration utilizing PEG-based hydrogels. To achieve the goal, the authors immunized C57BL/6 mice against PEG and performed a bone tissue engineering study in which calvarial defects are treated with bone morphogenetic protein-2 loaded PEG microporous annealed particle hydrogels to promote bone regeneration.

1. It is important that the authors use another set of non-PEG hydrogels loaded with bone morphogenetic protein-2. Otherwise, the data shown in Figure 2B are not convincing. There are many reasons that can cause the difference in data in Figure 2B

Thank you for this comment, but we disagree. Naïve mice which lack anti-PEG antibodies are the appropriate control for evaluating the effects of PEG sensitization. While we appreciate the reviewer's opinion, including a non-PEG hydrogel would complicate and potentially confound the study by introducing a new variable that is not relevant to the hypothesis we are testing.

2. Although the authors provide a lot of reasons to explain the difference, there are no direct correlations or evidences between anti-PEG antibodies generated and bone regeneration. At its form, it only reports what is observed with speculations. Without additional controlled experiments as described above, it is hard to make the connection.

Thank you for this comment. To strengthen our manuscript, we quantified the presence of CD3 positive T-cells and CD68 positive cells (macrophages and osteoclasts, which are potential sources of MMPs) and found significant differences in immune cell recruitment that may explain the differences. This new data is presented in **Figures 6**. While additional mechanistic studies are still needed, given the growing concerns over PEG immunogenicity, we believe it is appropriate to disseminate these results to the scientific community and follow up with mechanistic studies in future work. We hope the reviewer will agree.

3. Furthermore, the difference between male and female mice could be simply due to the difference from various possibilities such as a different level of anti-PEG antibodies generated, the different drug delivery profiles from these microgels and more.

We appreciate the reviewer's skepticism, but we designed our study in accordance with best practices on consideration of sex as a biological variable. We do not see how our finding of sex-dependent differences is scientifically invalid and kindly refer the reviewer to the Nature portfolio guidelines (<https://www.nature.com/nature-portfolio/editorial-policies/ethics-and-biosecurity#animal-research>) and the 'Sex and Gender Equity in Research – SAGER – guidelines' (DOI: 10.1186/s41073-016-0007-6).

4. Writing can be improved by paying attention to the detail. For example, there is no Figure 6 as cited in the text and Figure 2B caption should be better described.

Thank you for pointing this out. We have omitted the prior reference to Figure 6 in the text and revised the **Figure 2B** caption. We have also carefully checked the rest of the text for errors and incorrect figure citations.

Reviewers' Comments:

Reviewer #1:

Remarks to the Author:

The authors adequately addressed the prior comments raised.

Reviewer #2:

Remarks to the Author:

The manuscript was improved but it is unfortunate that more mechanistic studies could not be completed. I think it can be accepted with the changes already made.

Reviewer #3:

Remarks to the Author:

This is an important subject although mechanisms are still not fully understood. Authors answer most of my questions.

Ideally, a paper should have some mechanism studies rather than using data to explain what is going on. Realistically, it is hard for the authors to do anything significantly in term of mechanisms within a reasonable time frame based on their answers as these studies can be indeed complicated. Considering the importance of PEG immunogenicity, it may be OK to publish this paper to warn people about PEG immunogenicity for implants. It appears (also as claimed by the authors) that this is the first studies related to PEG immunogenicity and implants while most are on nanocarriers like many of our studies.